# QUIC is not Quick Enough over Fast Internet

Submission Id: 90

## ABSTRACT

QUIC is expected to be a game-changer in improving web application performance. In this paper, we conduct a systematic examination of QUIC's performance over high-speed networks. We find that over fast Internet, the UDP+QUIC+HTTP/3 stack suffers a data rate reduction of up to 45.2% compared to the TCP+TLS+HTTP/2 counterpart. Moreover, the performance gap between QUIC and HTTP/2 grows as the underlying bandwidth increases. We observe this issue on lightweight data transfer clients and major web browsers (Chrome, Edge, Firefox, Opera), on different hosts (desktop, mobile), and over diverse networks (wired broadband, cellular). It affects not only file transfers, but also various applications such as video streaming (up to 9.8% video bitrate reduction) and web browsing. Through rigorous packet trace analysis and kernel- and user-space profiling, we identify the root cause to be high *receiver-side* processing overhead, in particular, excessive data packets and QUIC's user-space ACKs. We make concrete recommendations for mitigating the observed performance issues.

## 1 INTRODUCTION

QUIC is a multiplexed transport-layer protocol over UDP, poised to be a foundational pillar of the next-generation Web infrastructures. It has recently been standardized by the IETF (known as *IETF QUIC* [48]) as the transport foundation of HTTP/3 [31]. Since 2013, QUIC has been commercially deployed by numerous companies including Google, Akamai, Meta, and Cloudflare [4, 9, 14, 22, 53]. As its adoption continues to grow rapidly, QUIC (together with HTTP/3) is standing at the forefront to reshape the performance paradigm of the World Wide Web, underpinning a multitude of applications and services. There is a plethora of literature on characterizing QUIC performance [42, 50, 56, 66, 71, 72, 79, 84]. They have used various QUIC implementations (customized vs. commercial), compute environments (mobile vs. desktop), and network conditions (wired vs. wireless). Due to such diversity, their findings are understandably a mixture of performance gains, and in some cases, degradations, compared to TCP and earlier generations of HTTP. In addition, a majority of these studies focus on low-throughput use cases.

In this study, we systematically examine an under-explored scenario: running QUIC over high-speed networks. This scenario is becoming increasingly important with the debut of faster networks such as high-speed wired links, WiFi 6/7, and 5G, which often reach more than 500 Mbps and up to 1+ Gbps per connection. Meanwhile, given the ubiquity of HTTP on today's Internet, HTTP (QUIC) is being utilized for bandwidth-intensive applications like ultra-high-resolution videos [63] and VR/AR [83]. This makes understanding QUIC's performance on high-speed networks even more crucial.

**QUIC is Slow over Fast Internet.** Despite typically being referred to as a transport-layer protocol, QUIC is deeply coupled with upper-layer components, namely TLS and HTTP. Its user-space nature makes such coupling more complex and extensive.

Note that an apple-to-apple comparison should be done on the UDP+QUIC+HTTP/3 protocol stack and TCP+TLS+HTTP/2 stack. For brevity, we refer to the two stacks as **QUIC** and **HTTP/2**. We begin with comparing QUIC and HTTP/2 in a simple environment: file download using a command-line data transfer tool, cURL [1], and a Chromium-based client, quic_client [26]. For a fair comparison, we keep factors such as the congestion control algorithm, server configuration, and network condition the same. The results show that QUIC and HTTP/2 exhibit similar performance when the network bandwidth is relatively low (below ~600 Mbps), whereas under a higher network bandwidth, QUIC consistently lags behind HTTP/2 by up to 15.7% in terms of throughput. The performance gap becomes more pronounced as the bandwidth increases. Notably, during packet reception, QUIC incurs considerably higher CPU usage than HTTP/2 on state-of-the-art client hosts.

Next, we investigate more realistic scenarios by conducting the same file download experiments on major browsers: Chrome, Edge, Firefox, and Opera. We observe that the performance gap is even larger than that in the cURL and quic_client experiments: on Chrome, QUIC begins to fall behind when the bandwidth exceeds ~500 Mbps. When the bandwidth reaches 1 Gbps, QUIC becomes 45.2% slower than HTTP/2. On weaker clients such as mobile devices, the gap is even larger.

**QUIC's Slowness Impacts Multiple Web Applications.** We experimentally demonstrate that QUIC's performance degradation affects not only bulk file transfers but also other applications including video content delivery and web browsing, despite their intermittent traffic patterns. QUIC incurs a video bitrate reduction of up to 9.8% compared to HTTP/2 when delivering DASH [73] video chunks over high-speed Ethernet and 5G. Again, such QoE degradation only exhibits when the underlying bandwidth is sufficiently high. For example, the impact is almost hidden over 4G but unleashed over 5G. QUIC's web page loading performance is less affected. Its page load time is 3.0% longer than HTTP/2's, averaged across 100 representative websites, with a long tail of page load time gaps over 50%.

**QUIC's Slowness over Fast Internet is due to Receiver-side Processing.** With the above results, we then identify the primary culprit of the QUIC-HTTP/2 performance gap. This is a highly challenging task due to a wide range of factors in the Web ecosystem, the high complexity of QUIC, and various engineering difficulties. We first make two observations by looking into packet traces and performance data: (1) The client running QUIC receives a much higher number of packets compared to those during HTTP/2 downloads; (2) There is a high delay between incoming data packets and their corresponding ACK packets when QUIC receives at a high data rate, suggesting that it takes longer to process QUIC packets. Both observations indicate that the slow performance of QUIC over fast Internet is due to limited receiver-side processing capability. It is important to note that although QUIC's user-space implementation is known to cause performance degradation in

general [53] and there have been efforts to optimize UDP/QUIC's sender-side transmission performance [7, 35, 46], we are *the first to identify the receiver side as a more likely performance bottleneck for QUIC over fast Internet.* This is not only because servers are typically more powerful than clients (desktops, laptops, mobile phones), but also attributed to unique challenges in handling data reception per QUIC's design, as detailed next.

**The Poor Receiver-side Performance is due to Excessive Data Packets and User-space ACKs.** We conduct deep performance profiling on the user-space Chromium (the open-sourced version of Chrome browser) and the underlying OS networking stack. We identify two main root causes of QUIC's poor receiver-side performance.

- **Issue 1.** When downloading the same file, the in-kernel UDP stack issues much more packet reads (`netif_receive_skb`) than TCP, leading to a significantly higher CPU usage. This is because none of the QUIC implementations we examine uses UDP generic receive offload (GRO) where the link layer module combines multiple received UDP datagrams into a mega data-gram before passing it to the transport layer. This is in sharp contrast to the wide deployment of TCP segmentation offload, and recent advocacy of UDP send-side offload (GSO).

- **Issue 2.** In the user space, QUIC incurs a higher overhead when processing received packets and generating responses. This over-head can be attributed to multiple factors: the excessive packets passed from the kernel (Issue 1), the user-space nature of QUIC ACKs, and the lack of certain optimizations such as delayed ACK in QUIC.

**Recommendations for Mitigation.** We make several recom-mendations for mitigating the above impact, including deploying UDP GRO on the receiver side, making generic offloading solutions (GSO and GRO) more QUIC-friendly, improving relevant QUIC logic on the receiver side, and using multiple CPU cores to receive data for QUIC. We also discuss some practical challenges of re-alizing the above recommendations, such as the heterogeneity of today's commodity client hosts (PCs, mobile devices, and embedded devices, with diverse OSes) compared to the servers.

At a high level, we advocate careful examinations of upper-layer protocols over emerging networks, applications, and services. This paper instantiates this idea by conducting a pioneering study on QUIC performance over high-speed Internet. We make two-fold contributions in this work: the measurement findings and the root cause analysis. We intend to release all the measurement data and source code of the study.

## 2 BACKGROUND AND MOTIVATION

QUIC is a user-space transport over UDP and comes with enforced encryption. It was initially proposed and developed by Google (**gQUIC**) [53] with the goal of enabling fast, reliable, and secure connections. Earlier, Google reported significant performance gains compared with TCP [5, 8]. An IETF working group was launched in 2016 to improve the original gQUIC design which fuses the trans-port, cryptographic handshakes, and upper-layer HTTP. They tease various functionalities into parts, and later standardized the refined version into **IETF QUIC** [48]. As the application layer wrapper of

**Table 1: Preliminary file download tests.**

| Testbed | Download Time (s) | | CPU Usage (%) | |
|---|---|---|---|---|
| | HTTP/2 | HTTP/3 | HTTP/2 | HTTP/3 |
| Desktop, Ethernet | 9.32 | 18.60 (+99%) | 77.5 | 96.9 |
| Pixel 5, low-band 5G | 37.11 | 78.65 (+112%) | 121.55 | 161.77 |
| Pixel 5, mmWave 5G | 30.10 | 63.20 (+110%) | 128.43 | 165.20 |

QUIC, HTTP/3 was also adopted as an IETF standard recently [31]. Essentially, HTTP/3 was structured to make the HTTP syntax as well as existing HTTP/2 functionalities compatible with QUIC. To-gether with the network layer and layers below, UDP, QUIC, and HTTP/3 form a new protocol stack for next-generation network communication, whose current counterpart is the stack of TCP, TLS, and HTTP/2. While QUIC's design brings benefits such as 0/1-RTT fast handshake, stream multiplexing for the removal of head-of-line blocking, and connection migration, there are also potential downsides. For example, QUIC involves processing and copying data between the kernel space and user space.

Downloading data over QUIC can become very slow in particular given the emergence of high-speed Internet. We conduct a prelim-inary experiment on both desktop and mobile Chrome browsers to download 1 GB files (see §3.1 for details). Table 1 presents the results averaged over 10 runs. We can find that, the file download time when QUIC is enabled is around double the time with QUIC disabled. The CPU usage is also higher during QUIC download. The performance disparity between QUIC and HTTP/2 is even larger on smartphones. Note that the CPU usage for the desktop is measured from the browser's network service while the measurement refers to the CPU usage of the entire browser process for the smartphone. CPU usage exceeding 100% indicates that the browser process was utilizing more than one cores in a multi-core system.

The results raise a couple of questions: When is QUIC data trans-fer slower than HTTP/2? What are the underlying reasons for the performance gap? Can users benefit from the current deployment of QUIC? To answer these questions, we carry out an in-depth mea-surement study on QUIC performance over high-speed networks.

## 3 QUIC TRANSPORT PERFORMANCE

In this section, we conduct a series of experiments comparing the performance of QUIC and HTTP/2. We start with introducing our experimental methodologies in §3.1. Then, we present file download experiments on lightweight data transfer clients in §3.2. Finally, we discuss the results on commercial web browsers in §3.3.

### 3.1 Methodology

Various factors within different components in the network can affect the overall performance and potentially become the bottle-neck. When comparing the UDP+QUIC+HTTP/3 (QUIC) stack with the TCP+TLS+HTTP/2 (HTTP/2) stack, we carefully set up the following testbed to ensure a fair comparison, that is, the observed performance gaps originate solely from the differences in the pro-tocol themselves.

We deploy a server machine equipped with an Intel Xeon E5-2640 CPU and a client desktop featuring an Intel Core i7-6700 CPU. They are connected through a 1-Gbps Ethernet, only two hops away

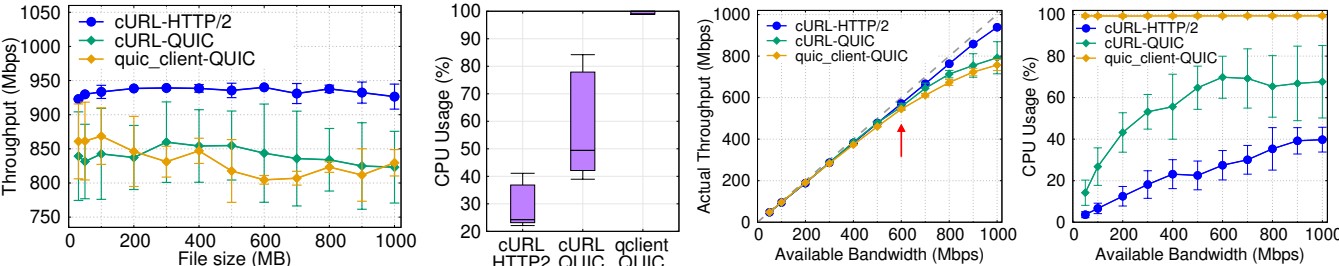

**Figure 1: Throughput of lightweight clients during file download.**

**Figure 2: CPU usage of lightweight clients.**

**Figure 3: Throughput and CPU usage of** `cURL` **and** `quic_client` **during file download under limited bandwidth.**

from each other. This setup avoids several network-related impacts such as network congestion and bandwidth throttling imposed by middleboxes which are often unfriendly to QUIC [38, 44, 53, 81]. Both machines run Ubuntu 18.04. We host an HTTP server using OpenLiteSpeed (v1.7.15) [25] built based on a mainstream QUIC library, LSQUIC [21]. The congestion control algorithm for QUIC is set to CUBIC, which is the default algorithm used for TCP in the OS. We also make sure their initial transport settings stay the same. Furthermore, both the UDP and TCP buffer sizes are adjusted to exceed 10x the link's bandwidth-delay product (BDP) to prevent buffer starvation during experiments. We run `tcpdump` to collect packet traces. We employ Linux `tc` [2] to control available network bandwidth when evaluating QUIC and HTTP/2 under low or changing bandwidth conditions.

## 3.2 File Download on Lightweight Clients

We start our investigation with a simplified setup, using two non-browser download tools, `cURL` [1] and `quic_client` [26]. `cURL` is a command-line data transfer tool that supports both QUIC and HTTP/2. `quic_client` is a standalone QUIC client implementation, built with the same QUIC stack as Chrome/Chromium.

We use the clients to download files of different sizes, ranging from 50 MB to 1 GB, over QUIC and HTTP/2. For each file size, both tools undergo 20 repeated download sessions. Figure 1 reports the mean values and standard deviations from the collected traces. The results show that `cURL` running HTTP/2 noticeably outperforms both QUIC clients, well utilizing the 1 Gbps available bandwidth. `quic_client`'s results stand very close to those of `cURL` on QUIC. On average, the throughput of `cURL` running QUIC and that of `quic_client` is 7-16% and 8-12% lower, respectively, compared to `cURL` with HTTP/2. Moreover, both `cURL` on QUIC and `quic_client` display an almost parallel trajectory, which indicate the similar efficiency in their QUIC implementations.

We present in Figure 2 the distribution of the client's CPU usage during the download of a 1 GB file. The CPU usage for `cURL` when running QUIC is higher than that of `cURL` on HTTP/2. `quic_client`'s CPU usage is further elevated, nearly maxing out at 100%, while its throughput remains similar to `cURL` on QUIC. Note that, for `quic_client`, we have deactivated any debug mode for optimal performance (`is_debug=false`). Since it is a simplistic implementation of the QUIC protocol stack, for instance, not designed

for handling multiple concurrent connections or non-transfer functionalities such as logging, it just consumes all available CPU resources during the download process without reservation, unlike `cURL` which is engineered for versatility across various scenarios.

We next limit the available network bandwidth from 50 Mbps to 1000 Mbps. As shown in Figure 3, when the available bandwidth is low, QUIC and HTTP/2 exhibit similar performance. Both QUIC clients can catch up with the available bandwidth, with `quic_client`'s throughput being slightly lower. However, as the bandwidth provision grows beyond around 600 Mbps, QUIC's actual throughput starts to be bottlenecked and a noticeable throughput disparity between QUIC and HTTP/2 emerges. The CPU usage for `quic_client` is always high and that of `cURL` QUIC hovers around 70%, reemphasizing the computational challenges associated with the protocol. We analyze possible performance inhibitors leading to the high CPU usage later in §5.

## 3.3 File Download on Real Browsers

Transitioning from lightweight clients, we look into experiments on real web browsers. This exploration mainly focuses on the well-known Chrome browser.

We repeat the file download tests on Chrome. As shown in Figure 4, the performance gap between QUIC and HTTP/2 is even larger than that in our prior lightweight client experiments (§3.2). Figure 5 plots the CPU usage of the network process ("Utility: Network Service" [27], responsible for network-related tasks.) during the download. It is evident that the Chrome browser running QUIC demands more computational power than Chrome with HTTP/2. Different from the lightweight `cURL` and `quic_client`, Chrome is a full-fledged web browser, so the CPU saturation issue is exacerbated, leading to even lower QUIC performance. Remarkably, QUIC's average throughput can barely hit 478 Mbps.

The experimental results in controlled bandwidth scenarios are depicted in Figure 6. QUIC fails to fully utilize the bandwidth starting earlier at approximately 500 Mbps, compared to the 600 Mbps bottleneck point identified in the lightweight client tests (see Figure 3). Chrome with QUIC approaches 100% CPU usage when the throughput is only 200 Mbps. Recall that, with further limited compute resources, the HTTP/2-QUIC performance gap on mobile devices is more pronounced, as shown in Table 1.

Additionally, we run experiments of changing the CPU frequency (*i.e.,* CPU clock speed). The Intel Core i7-6700 CPU equipped on the client machine has a base frequency of 3.40 GHz and can be boosted to 4.00 GHz. In Figure 7, as we reduce the CPU frequency, Chrome's

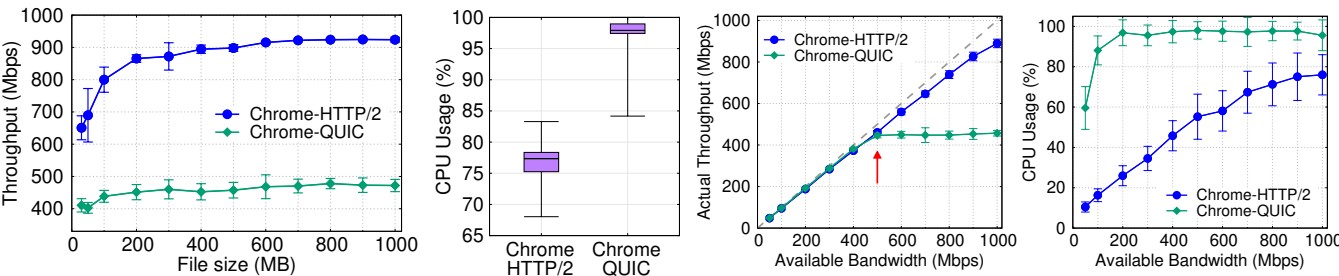

**Figure 4: Throughput of the Chrome browser during file download.**

**Figure 5: CPU usage of the Chrome browser.**

**Figure 6: Throughput and CPU usage of the Chrome browser during file download under limited bandwidth.**

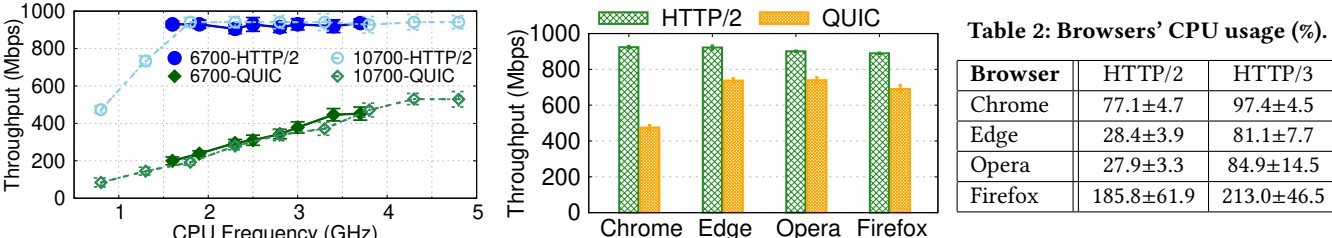

**Figure 7: Throughput of the Chrome browser at different CPU frequencies.**

**Figure 8: Throughput of four different browsers during file download.**

**Table 2: Browsers' CPU usage (%).**

| **Browser** | HTTP/2 | HTTP/3 |
|---|---|---|
| Chrome | 77.1±4.7 | 97.4±4.5 |
| Edge | 28.4±3.9 | 81.1±7.7 |
| Opera | 27.9±3.3 | 84.9±14.5 |
| Firefox | 185.8±61.9 | 213.0±46.5 |

QUIC download throughput further drops to around 200 Mbps while the throughput over HTTP/2 still remains above 900 Mbps even at 1.60 GHz. Note, unless otherwise specified, all the other experiments in this work are done with the CPU set to 3.40 GHz. Then, we test on a machine with a more advanced CPU, Intel Core i7-10700 with a 4.80 GHz maximum turbo frequency. The QUIC downlink throughput stays close compared to the 6700 machine at the same frequency while it can reach 530 Mbps at 4.80 GHz. This suggests that increasing CPU computing power can marginally narrow the performance gap between QUIC and HTTP/2.

**Comparing Different Browsers.** In addition to Google Chrome (v102), we extend our HTTP file download experiments to other QUIC-enabled web browsers: Mozilla Firefox (v105), Microsoft Edge (v106), and Opera (v93). We plot the download throughput statistics for four browsers in Figure 8 and list their CPU usage data in Table 2. Note that we were unable to isolate the CPU usage of Firefox's network service but we ensure that no other activities running in Firefox. We find that, all the browsers have a worse performance when QUIC is enabled, with increased CPU usage. Therefore, the slow QUIC download issue is prevalent across major commercial browsers. This can significantly affects the end-user experience, especially when downloading bulk data at a high speed.

# 4 APPLICATION STUDY

Our experimental findings have painted a compelling narrative about QUIC's performance not just in bulk file transfers, but also other applications, including video content delivery and web page loading, despite their intermittent traffic patterns. In this section, we delve into these specific application areas to further showcase the impact of QUIC.

## 4.1 Video Streaming

The vast and growing demand for high-quality video content and smooth delivery on the Internet underscores the importance of efficient protocols for adaptive bitrate (ABR) video streaming [28]. Leveraging both QUIC and HTTP/2, we set out to explore their real-world implications on video streaming performance.

We employ ffmpeg to encode a custom 4K video with H.264, generating six tracks at different bitrates. 4K video streaming usually requires 35-100 Mbps [11, 18, 60], which can be easily achieved by today's high-speed networking like 5G. In order to challenge the rate adaptation controllers, avoid trivial bitrate selection, and examine future ultra-high resolution videos and extended reality (XR) performance over high-speed connectivity, we scale up the video bitrates with the top track bitrate reaching 200 Mbps, to match the median throughput of 5G network traces [60]. Specifically, the bitrates are 20 Mbps, 40 Mbps, 80 Mbps, 120 Mbps, 160 Mbps, and 200 Mbps. We also encode the video into three different chunk durations, 1s, 2s, and 4s. We set up a dash.js server for ABR video streaming. The server is configured to support two major categories of bitrate adaptation algorithms: Buffer-Based (BB) [45] which selects bitrates with the goal of keeping the buffer occupancy high, and Rate-Based (RB) [55] which selects the highest bitrate below the bandwidth predicted from experienced throughputs during past chunk downloads. The client machine runs a Chrome browser to fetch and play the video content from the server.

We evaluate ABR video streaming under three types of network conditions. In addition to the 1 Gbps Ethernet link considered in our previous experiments, we also run tc [2] to emulate 4G and 5G networks using real network traces, randomly selected from the Lumos5G dataset [59]. For each network type, we have two traces each for walking and driving scenarios to incorporate various mobility patterns. We conduct such a trace-driven emulation to

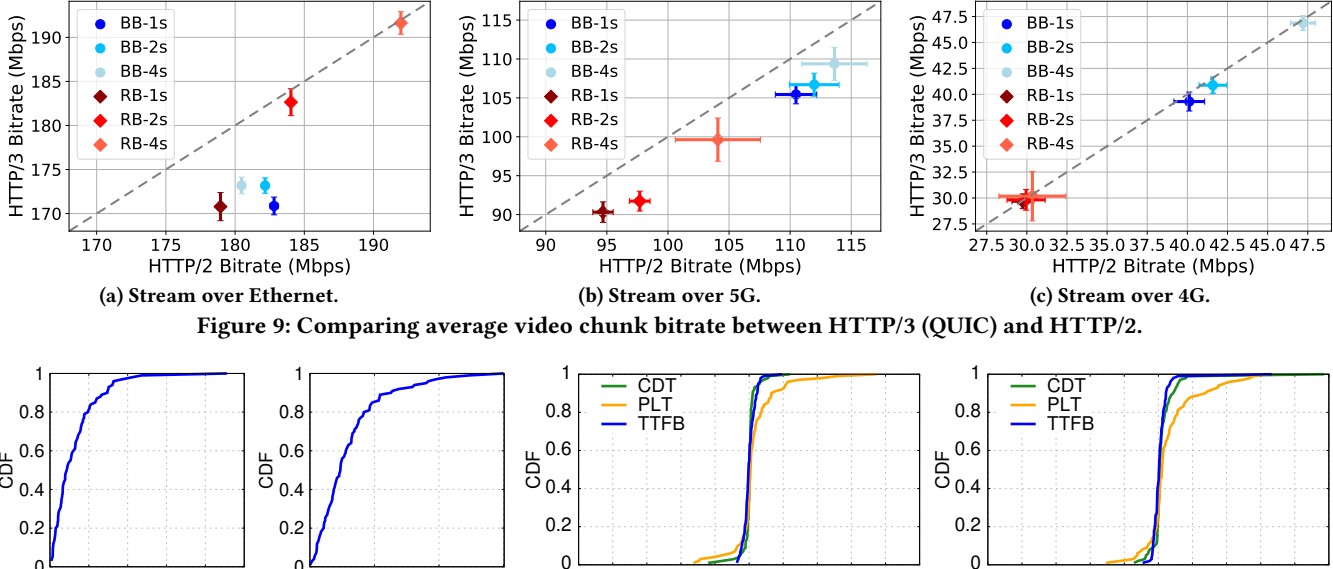

(a) Stream over Ethernet.      (b) Stream over 5G.      (c) Stream over 4G.

Figure 9: Comparing average video chunk bitrate between HTTP/3 (QUIC) and HTTP/2.

Figure 10: Website characterization.          Figure 11: Web page loading results (HTTP/3 over HTTP/2).

ensure QUIC and HTTP/2 experience the same set of network conditions and to provide better reproducibility across different rounds.

We measure video chunk bitrate and CPU usage during the streaming process. Each experimental setup is executed 20 times. As shown in Figure 9, the results of streaming ABR videos over QUIC and HTTP/2 suggest that QUIC performs worse than HTTP/2 in Ethernet and 5G scenarios. The bitrate reduction goes up to 9.8%. This is likely due to the bandwidth in these two network settings being high enough to saturate the client CPU. Revisiting our earlier discussions in §3, we discover that, the bottleneck bandwidths after which QUIC cannot fully utilize the link capacity for the lightweight clients and Chrome are around 500 Mbps and 600 Mbps, respectively. Taking into account the video playback overhead (*e.g.,* decoding and rendering), this bottleneck point could be further lowered. On the other hand, for the slow 4G networks, shown in Figure 9c, the performance difference is not that significant. The HTTP/2 setups have a slightly better overall bitrate.

## 4.2 Web Page Loading

Web browsing (*i.e.,* web page loading) plays another crucial role in the Web ecosystem. Unlike bulk file download, loading a web page usually involves transferring multiple small objects that can be either concurrent or sequential depending on object dependencies. We conduct an extensive experiment with Alexa's top 100 websites.

First, we use the original URLs to directly load the remote websites, with QUIC enabled on Chrome. We repeat the tests on each website 20 times. Surprisingly, the page load tests on most websites do not capture any HTTP/3 objects, which means those websites have not enabled QUIC yet. Only 16 websites exhibit HTTP/3 traffic during page loads. The website containing the largest portion of HTTP/3 objects is www.discord.com with no HTTP/1.1 objects, 8.5 HTTP/2 objects, and 30.0 HTTP/3 objects on average. It is also

noteworthy that, with various third-party links (for example, for tracking or generating dynamic content purposes) visited from JavaScript files embedded in the main page, none of the tested websites are completely loaded over HTTP/3.

Then we download these 100 websites using SiteSucker [23] and host them locally on our web server. One challenge we encountered is, to our knowledge, there is no tool that can be used to download an entire website with countless external links so that it can be fully hosted locally. There are tools to record web page load and replay locally, *e.g.,* Web Page Replay (WprGo) [19] and Fiddler [20]. However, since they do not really copy the entire website, it is not possible to toggle between different HTTP protocols during the replay. Figure 10 shows the website features which include the number of objects and the total page size, illustrating the diversity of our test websites. Using chrome-har-capturer [12], we build scripts to collect HTTP Archive (HAR) [3] files and calculate the evaluation metrics. To compare the page load performance of QUIC and HTTP/2, we utilize three major metrics: (1) content download time (CDT) [33], defined as the time to download all content needed to load the website, after which the rendering process can start; (2) page load time (PLT) [62, 77], at which the rendering of all components of the page is finished; and (3) time-to-first-byte (TTFB) [32], which is the delay from sending the request to receiving the first byte of the response.

We repeat the page load tests 20 times for each website over Ethernet. We also use tc to throttle bandwidth at 100 Mbps, to examine the performance at limited bandwidth conditions (*e.g.,* 4G/5G). Note that we do not directly use mobile network traces because a web page load is too fast, making it difficult to ensure consistent network conditions across rounds by replaying traces. Figure 11 compares the timers (CDT, PLT, and TTFB) for QUIC and HTTP/2. A data point greater than 1.0 means the corresponding timer is longer in QUIC tests. We can learn from the results that,

the performance difference is not as significant as that observed in video streaming tests. On average, QUIC's PLT is 3.0% longer than HTTP/2's. However, there is a long tail indicating that in some cases the gap can be over 50% and up to 74.9%. We also observe the increased CPU usage in QUIC, compared to HTTP/2 page load tests. The PLT increase is not as significant as the bulk download time increase, because web page loading involves both local page rendering, which is not affected by the network protocol selection, and network data transfer.

## 5 ROOT CAUSE ANALYSIS

With the QUIC and HTTP/2 results on various applications, we now identify the root cause of the observed performance gap. Unless otherwise noted, for experiments and analysis in this section, we use Chromium (v102) as it is a production-level implementation supporting both HTTP/2, and QUIC and it is not proprietary, thus easy to profile internal activities.

### 5.1 Eliminating Non-contributing Factors

We begin with eliminating several potential factors, most backed up with controlled experiments.

- **Server Software.** We set up another web server, Nginx-quic (v1.14.0) [24], on the same server machine. We compare the time to download 1 GB file from Nginx and from OpenLiteSpeed (our server setup in §3) using Google Chrome. HTTP/2 performs similarly on both web servers while QUIC performs even worse when running on Nginx, being slower by 18%.

- **UDP/TCP Protocols.** We conduct `iPerf` UDP and TCP tests under the same network setup. The results show that both protocols can fully utilize the link bandwidth (1 Gbps), with UDP achieving 958 Mbps and TCP achieving 944 Mbps on average.

- **HTTP syntax.** HTTP/3 [31] serves as the mapping of HTTP for using QUIC as the transport. Adapted from HTTP/2 [30], it has an almost identical syntax structure to HTTP/2 [13, 41].

- **TLS Encryption.** Both QUIC (TLS v1.3) and HTTP/2 (TLS v1.2) employ the TLS_AES_128_GCM_SHA256 cipher on our web server. We also benchmark different cipher suites and the results do not significantly affect the performance.

- **Parameter Tuning.** We tune QUIC-specific parameters such as enabling/disabling packet pacing and adjusting path MTU discovery [40]. We do not observe noticeable improvements compared to the original performance gap.

- **Client OS.** We repeat the above experiments on Mac OS and Windows for the receiver, and observe similar results.

- **Disk and Memory.** We download files directly to a volatile RAM-based disk using Linux `tmpfs` [17]. We also test with Linux HugePages [29] to avoid frequent memory swaps. Neither approach helps in improving QUIC performance.

### 5.2 Evidence from Packet Trace Analyses

Next, We get insights by analyzing `tcpdump` packet traces.

**QUIC Perceives Much More Packets than HTTP/2.** We notice that for QUIC, the number of packets received by the OS's UDP stack is an order of magnitude higher than the number of packets received by the TCP stack during HTTP/2 downloads (744K versus 58K on average). We have confirmed this is not caused by retransmissions. While prior tests show that increasing the packet size up to MTU can help [10], all QUIC packets in our experiments are already MTU-sized (1472 bytes, excluding the 8-byte UDP header and the 20-byte IP header in the standard 1500-byte MTU setup). We also verify that the numbers of packets transmitted over the wire are very close between QUIC and HTTP/2. The difference of their transport-layer-perceived packets is because TCP (HTTP/2) uses generic receive offload (GRO), where the link layer module in the OS combines multiple received TCP segments into a large segment of up to 64 KB. However, despite the availability of UDP GRO, it is not used by QUIC, and integrating GRO with QUIC faces challenges as to be discussed in §5.3.

**QUIC has a much Higher RTT Dominated by Local Processing.** We measure the packet round-trip time (RTT), defined as the time between when a data packet is sent out from the server and when the first packet to acknowledge it is received. The RTT consists of the propagation delay spent on the paths and the processing delay spent on the receiver side. Though TCP and QUIC have different ACK mechanisms, the average packet RTT can still reflect how fast packets are transferred and processed, and thus help adjust the sending rate. The average RTT for HTTP/2 download is 1.9ms while QUIC's RTT skyrockets to 16.2ms. Since the `ping` RTT between the two machines is only 0.23ms as measured, the endpoint packet processing takes most of the packet latency.

The above results provide further evidence that the performance bottleneck of QUIC appears to be on the receiver side.

### 5.3 Root Causes via OS/Chromium Profiling

To definitively pinpoint the root cause, we conduct fine-grained profiling in both the OS kernel (OS's networking stack) and the user space (Chromium's networking stack) using Linux `perf` [16].

**Excessive Receiver-side Processing in the Kernel.** We run 1 GB file downloads on Chromium with QUIC and HTTP/2. Meanwhile, we use `perf` to monitor events in the Linux networking subsystem (`net`) associated with Chromium's network service. For QUIC, we observe a huge number of calls on `netif_receive_skb` which is invoked when a packet is received at the network interface. Specifically, there are 231K calls of this type witnessed during a single QUIC download compared to a mere 15K in an HTTP/2 download. This difference roughly corresponds to the difference in the number of received UDP and TCP packets (§5.2).

A standard way to reduce packet processing overhead in the OS is to involve NIC offloading that has been widely used for TCP, including segmentation offload such as TCP Segmentation Offload (TSO) and Generic Segment Offload (GSO) on the sender side and receive offload such as Generic Receive Offload (GRO) on the receiver side[1]. While some existing efforts [7, 10] have shown the effectiveness of UDP sender-side offloading, our work pioneers in pinpointing the criticality of *receiver-side offloading* for today's commodity QUIC client hosts.

---

[1]Another solution, UDP Fragmentation Offload (UFO), uses IP fragmentation. It was deprecated so we do not consider it in this work.

**Table 3: Download 1 GB file with and without offloading.**

| Setup | # Sent Packets | # Recv Packets | Time (s) |
|---|---|---|---|
| QUIC (on) | 743K | 743K | 18.60 |
| QUIC (off) | 744K | 744K | 18.82 |
| HTTP/2 (on) | 19K | 53K | 9.36 |
| HTTP/2 (off) | 744K | 744K | 10.84 |

**Table 4: A breakdown of packet processing time.**

| Chromium Networking Stack | QUIC (8.5s) | HTTP/2 (4.1s) |
|---|---|---|
| Read UDP/TCP packets from socket | 0.248s | 0.037s |
| Process UDP/TCP packets for payload | 0.310s | 0.084s |
| Decode QUIC/TLS-encrypted packets | 0.660s | 0.814s |
| Parse decrypted QUIC/HTTP2 frames | 3.468s | 3.182s |
| Generate QUIC responses (*e.g.*, ACK) | 2.972s | – |
| Others | 0.859s | 0.001s |

In addition, we note that realizing offloading for QUIC is challenging. First, unlike TCP which uses a byte stream model so its payload can be flexibly (re)packetized, UDP's offloading logic must preserve the packet boundaries. The existing UDP GSO/GRO thus only supports offloading a train of UDP packets with identical lengths specified by the application [35]. This constraint makes directly applying UDP GSO/GRO to QUIC inefficient, due to QUIC's inherent multiplex nature: QUIC frames belonging to different streams vary in size and are multiplexed after encryption. As a result, if a train of UDP datagrams (containing the encrypted frames) have different packet sizes, existing UDP GSO/GRO cannot offload them. Second, blindly aggregating many UDP datagrams and transmitting them in a single burst may cause congestion-related packet losses and fairness issues, particularly over the wide-area Internet [35, 46]. Third, the diverse QUIC variants add complexity to realizing the QUIC offloading logic in NIC hardware. Likely due to the above reasons, although UDP GSO/GRO [6] is available in the newer Linux kernel versions, to our knowledge none of the QUIC implementations have adopted it.

We carry out additional experiments with available offloading mechanisms (TSO, GSO, and GRO) enabled and disabled on both server and client sides. The results in Table 3 indicate that UDP (QUIC) does not benefit from GRO/GSO. In contrast, TCP shows a more significant reduction in download time, with much fewer packets processed by the OS's TCP stack. The discrepancy in the number of packets sent and received is likely because the server-side offload may have a different power on segmentation compared to client-side receive offload capability on packet reassembly. Note, all other experiments in this study have them turned on.

When profiling kernel-level activities for QUIC, we also observe a more significant proportion of calls to function do_syscall_64 (17K for QUIC, compared to 4K for HTTP/2) and function copy_user_enhanced_fast_string (4K vs. 3K). Such intensive interactions across the user-kernel boundary are resulted from the substantial volume of QUIC packets perceived by the UDP stack. They further increase the processing overhead.

**Excessive Receiver-side Processing in the User Space.** The high in-kernel packet processing overhead results in high processing overhead in the user space for QUIC. To demonstrate the latter, we profile Chromium's networking stack, specifically, the

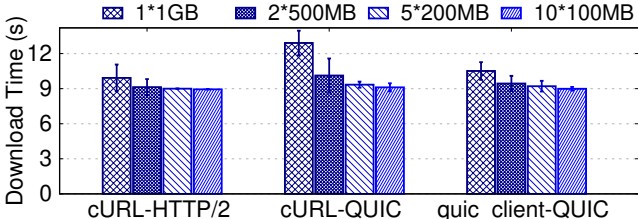

**Figure 12: Parallel download experiments (instances of** cURL **or** quic_client **download 1 GB of files in total).**

Chrome_ChildIOT thread. Table 4 provides a breakdown of the time spent by each packet processing stages. The stack is primarily responsible for (1) reading UDP/TCP packets from the socket; (2) processing UDP/TCP packets to extract the payload; (3) decoding QUIC/TLS-encrypted packets; and (4) parsing decrypted QUIC/HTTP2 frames. For QUIC, QuicChromiumPacketReader is responsible for reading and processing the incoming QUIC packets. Its entry point is StartReading and consumes 8.7s out of the total download time of 20.6s on average when downloading a 1 GB file. On the flip side, the HTTP/2 counterpart is SpdySession starting from DoReadLoop, and spends 4.1s out of 9.4s. QUIC lags behind HTTP/2 at each of the four stages above. Furthermore, we notice that, out of the 8.7s consumed by QuicChromiumPacketReader, QUIC spends 3.0s generating responses such as ACKs. In contrast, for HTTP/2, the ACKs are handled by the OS kernel, and they are generated more efficiently and sparsely due to various optimizations such as TCP delayed ACK and receive offload.

## 6 RECOMMENDATIONS FOR MITIGATION

Following the above experiments and analysis, we make several recommendations for mitigating the observed issues.

**Adoption of UDP GRO on the Receiver Side.** Most importantly, UDP GRO needs to be deployed on the receiver side to reduce the number of packets handled by the UDP stack. This will reduce not only the in-kernel overhead, but also QUIC's processing overhead in the user space. However, given the heterogeneity of today's commodity hosts (PCs, mobile devices, and even embedded devices, with diverse OSes), wide deployment of UDP GRO can be challenging, not to mention supporting it in the NIC hardware.

**QUIC-friendly Improvements to the offloading solutions.** We advocate that the generic offloading solutions (GSO and GRO) need some QUIC-friendly improvements. First, UDP GSO/GRO needs to support offloading a train of packets with different sizes (§5.3). Second, UDP GSO needs proper pacing configurations (*i.e.,* avoid transmitting too many UDP packets in a single burst that may incur network congestion) over the wild Internet. Ideally, the pacing logic should be properly interfaced with QUIC's logic such as congestion control[2].

**Optimizing QUIC logic on the Receiver Side.** There is also room for improvement in the QUIC logic on the receiver side. Sending delayed QUIC ACKs [47] can help reduce the overhead on generating QUIC responses. Besides, we note that Chromium currently uses recvmsg to read individual UDP packets; using recvmmsg to

---

[2]Google has a simple experimental pacing design for UDP GSO, but it is not designed specifically for QUIC and was only tested in data center networks.

read multiple UDP packets in a single system call may help improve the receiver-side performance.

**Multi-threaded download.** we also notice that Chromium uses a single thread for receiving network data. When fetching large files, using multi-threaded download (each thread running on a separate CPU core) can improve the receive-side performance. Since the tested Chrome browser version does not have built-in support of multi-threaded download, we conduct an experiment where we launch $k$ instances of cURL or quic_client, each downloading a file of 1 GB/$k$ ($k = 1, 2, 5, 10$). We use the latest finishing time across the $k$ instances to calculate the overall transfer time of 1 GB worth of data. As shown in Figure 12, increasing $k$ helps reduce the download time, in particular for QUIC. Nevertheless, similar to parallel TCP connections, this approach may incur fairness issues in network resource allocation. The sender could use existing solutions such as coupled congestion control [68] to bound the aggregated aggressiveness of the $k$ QUIC sessions.

## 7 RELATED WORK

We discuss existing literature related to QUIC, its performance measurements, industry-driven optimizations, and other relevant works. Then we highlight how our study differs from them.

**QUIC Measurements.** Since its advent at Google in 2013, QUIC has been extensively researched in the literature. Google presented their experience with QUIC after years of Internet-wide deployment. Carlucci *et al.* [34] examined an early version of QUIC (v21) and showed its superior performance over TCP. Likewise, Megyesi *et al.* [57] compared QUIC with SPDY [78] and HTTP/1.1 and highlighted QUIC's performance improvements. QUIC's rapid evolution has led to efforts investigating the interoperability across QUIC implementations [49, 56]. Longitudinal studies, such as by Kakhki *et al.* [50] and Piraux *et al.* [66], traced QUIC's progression over time. Rüth *et al.* [70] took a first look at QUIC deployment and usage at an early stage. Similarly, QScanner [86] is implemented to analyze early QUIC deployments. There is also a rich tapestry of research diving into the impact of QUIC on various applications including videos streaming [64, 80], web browsing [71, 79], and a mix of different workloads [72, 84], and on various platforms including mobile [42] and satellite communication [51]. None of these works specifically focus on IETF QUIC's performance on major applications over high-speed networks as we do.

**QUIC Performance Optimization by Industry.** With various measurement studies on different aspects of QUIC, some efforts have been made to optimize QUIC from the industry. Through presentations and blogs, companies like Google [46], Cloudflare [7], and Fastly [10] reported their progress on optimizing QUIC. Some of their findings such as using UDP GSO [6] are relevant to ours. However, all the above works are concerned with the server-side performance. Since downlink traffic dominates the overall server-side traffic, they naturally focus on optimizing data transmission performance. Furthermore, some of their measurement methodologies are not very realistic or not documented in details, making it difficult to reproduce the experiments. For example, Cloudflare [7] made some attempts in accelerating sending data over QUIC including using sendmmsg and UDP GSO, but both the client and server

run on the same host (a laptop) instead of a production environment. Fastly [10] attributed the high computational cost of QUIC to ACK processing and per-packet sender overhead. However, they also focus on the sender-side optimization and use a simple setting where the CPU's clock is limited at only 400 MHz in order to measure the throughput sustainable with all available computational power. Red Hat [15] also mentioned enabling optional TSO/GSO support in a new release but omitted a performance examination on QUIC. In contrast, we investigate the receiver-side performance bottleneck through rigorous measurements and profiling on multiple dimensions (network traces, OS kernel, QUIC runtime, and higher-layer applications). Besides, we identify new challenges in making offloading solutions harmonize with QUIC.

**Other Works on QUIC.** Some prior works extended QUIC to multipath QUIC (MPQUIC) [36]. Researchers have further followed up with new packet schedulers for MPQUIC [63, 67, 76] and upper-layer solutions [85]. MultipathTester [37] is an iOS application for comparing the performance of MPTCP with MPQUIC. Several studies focused on QUIC congestion control [39, 43, 58] and some others look at the security and privacy [52, 54, 61, 74] aspects of QUIC. eQUIC Gateway [65] is a kernel module for accelerating QUIC through eBPF XDP. It is proposed to offload QUIC logic to NIC [82]. There are also new experimental protocols motivated by QUIC, such as TCPLS [69] and DCQUIC [75]. Our work on the root causing of QUIC's slowness offers unique contributions that set it apart from all the above studies.

## 8 CONCLUDING REMARKS

QUIC, with its design principles aimed at eliminating the head-of-line blocking problem, introducing fast connection establishments, and integrating transport-layer security, promises a more responsive and secure Web experience. However, our in-depth study, along with others, has highlighted areas where QUIC might not meet expectations. Our extensive measurements have underscored that, in some environments such as fast Internet (>500 Mbps in our experiments), QUIC's performance may not always live up to its name ("quick"). Through comprehensive performance profiling, we reveal the root cause to be the pronounced *receiver-side* processing overhead. This overhead manifests in the form of excessive data packets observed at Layer 3 and above, as well as QUIC's distinctive user-space ACKs.

There are notable challenges to grapple with. The absence of certain offloading techniques like UDP GRO, the user-space nature of QUIC, and QUIC's inherent reliance on UDP might complicate its deployment, especially in environments that have been meticulously optimized for TCP. Nevertheless, it is pivotal to note that QUIC is still in its nascent phase, with considerable research, exploration, and development fervently aiming to enhance its performance. The ongoing efforts and collaborations from multiple stakeholders in the Web ecosystem, including OS vendors, QUIC developers, and standardization organizations, will play a crucial role in the evolution of QUIC. As more web services transition to HTTP/3, we can expect a broader adoption of QUIC across the Internet. We hope that our findings can spur more explorations to improve QUIC, and upper-layer protocols in general, boosting their performance for the next generation networks, services, and applications.

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
