# OpenReview forum: "QUIC is not Quick Enough over Fast Internet"
_ACM.org/TheWebConf/2024/Conference — TheWebConf24 Oral_

### Official Review · Reviewer_cmo7 · 2023-11-17

**Novelty:** 5
**Technical Quality:** 5

**Review:**

This paper finds that the QUIC protocol performs lower than HTTP/2 over fast Internet through extensive testing on different web browsers, hosts, and network scenarios. By evaluation with LSQUIC, it is found that the reason is due to the high processing overhead in the QUIC's receiver-side. It is an interesting paper and I think the observation is useful.

Strengths：

- Discover a significant and important fact that QUIC's performance is not as good as HTTP/2 over fast Internet.

- Conduct experimental tests on different browsers (Chrome, Edge,Firefox), hosts (desktop, mobile), and network scenarios (wired broadband, cellular), making the results more convincing and robust.

- Use practical application scenarios such as video streaming and web page loading, rather than limited to file downloads. Similar results were also observed in video streaming transmission.

Major weaknesses:

- Lacks of measurement for the influence caused by the QUIC special mechanism such as the stream control, multiplexing and FEC.

**Questions:**

This article focuses on the performance differences between QUIC and HTTP/2 in fast networks, exploring the phenomenon and reasons for poor performance of QUIC in practical applications such as file download, video streaming, and web page loading in high-speed networks. All of these points are novel and have practical discussion value and significance. There are some limitations for this paper, and here are some comments which perhaps can be discussed further.

1. In figure 3, it's a bit strange that quic_client-QUIC's CPU usage is always at full capacity from start to finish. Can you explain it in detail?

2. The results in Figures 4 and 5 suggest that the throughput is limited due to insufficient CPU resources. Of course, this is mainly due to the high workload on the client side, which has already been mentioned in the article. Is it possible to further improve the performance of QUIC with sufficient CPU resources?

3. I think analysis for the QUIC architecture and comparison it with TCP/HTTP2 architecture at background section can help the reader to understand QUIC better. And that will also make this paper stronger.

4. Lacks of measurement for the influence caused by the QUIC special mechanism such as the stream control, multiplexing and FEC. More measurements about them are attractive for readers.

5. How to measure the CPU usage in Figure 6? Using the Linux command 'top'?

6. In network measurement, the size of packet has great impact on the CPU usage and file transmit speed. This paper lacks discussion of different packet sizes in the experiments.

**Reviewer Confidence:**

4: The reviewer is certain that the evaluation is correct and very familiar with the relevant literature

**Scope:**

4: The work is relevant to the Web and to the track, and is of broad interest to the community

---

### Official Review · Reviewer_r9sF · 2023-11-22

**Novelty:** 7
**Technical Quality:** 6

**Review:**

This paper proposes a thorough study of QUIC performance over the fast Internet. The authors have made a setup of QUIC server (LSQUIC) and QUIC client under different bandwidth patterns and over different applications, ranging from the stand-alone applications to browser-based applications, as well as video streaming. The authors have also updated system-level parameters like the clock frequency, turbo frequency, etc. to analyze the impacts of system parameters on QUIC. From a thorough analysis, the authors have concluded that QUIC does not perform well over fast Internet, and the reason behind that is the receiver-side processing overhead of user-space connection handling and congestion control mechanism on top of UDP.

**Quality:**

The overall quality of the paper is quite good. The authors have provided a methodological details about the data collection and analysis of the protocols at very granular level. I liked the root-cause analysis presented in the paper.

**Clarity:**

The clarity of the paper is good. The presentation is also nice and easy to follow.

**Originality:**

Although there have been a large number of papers on QUIC performance analysis, this paper provides a new insight of QUIC performance by analyzing the receiver-side processing overhead of the user-space events.

**Significance:**

The work is significant for the web community.

Overall I liked the paper and the analysis presented in this paper. Indeed, I must say that we have observed some of the phenomena reported in this paper during our own works on QUIC analysis; however, the authors have analyzed the observations very systematically and reported them in the paper methodically. I also liked Section 6 in the paper which nicely provides the take-away messages from such a measurement paper. Notably, the authors could have extended their experiments with multiple other possible setups, like impact of FEC, Multi-path QUIC, stream multiplexing, etc.; however, my personal view is that the content of this paper is good enough for publication and will be interesting for the web community.

**Questions:**

1) Figure 8 is interesting. It shows that HTTP/2 performance was almost similar for all the browsers, but QUIC performance depends significantly depending on the browser used. Is it something related to the browser processing overhead, or something else?

2) The authors have reported that QUIC has a much higher RTT -- but is the temporal RTT patterns same or different between HTTP/2 and QUIC? If they are different, that means the same congestion control (Cubic) might perform differently for HTTP/2 and QUIC for the same network condition. This can be very interesting from the protocol design performance -- the authors might look into that.

3) What is the packet size distribution that you observe for QUIC at the received side? That might have an impact of the receiver-side processing overhead.

**Reviewer Confidence:**

4: The reviewer is certain that the evaluation is correct and very familiar with the relevant literature

**Scope:**

4: The work is relevant to the Web and to the track, and is of broad interest to the community

---

### Official Review · Reviewer_fz43 · 2023-11-24

**Novelty:** 5
**Technical Quality:** 5

**Review:**

## Summary

HTTP/3 (and QUIC that underpins it) is a relatively new Web protocol, that has the potential to improve application performance in a range of scenarios. While the performance of QUIC and HTTP/3 has been well-studied across a range of implementations and network scenarios, there is limited characterisation across low-latency, high bandwidth connections. As such connections become more commonplace, it becomes increasingly important to consider performance on these networks. This paper presents a characterisation of QUIC and HTTP/3 performance across fast Internet links. The evaluation is comprehensive, with results from lightweight clients (like cURL), web browsers (Chrome, Edge, ..), and with different applications (including Internet video and web browsing). Through a root cause analysis, the paper concludes that high receiver-side processing overheads in QUIC result in significant performance gaps between the HTTP/3 and HTTP/2 stacks.

## Reasons to accept
- The paper is well-written, and the evaluations are comprehensive, and well conducted.
- The literature has studied the sender-side processing overheads well, but this work does a good job of highlighting the receiver-side overheads, that result from both QUIC's use of UDP, and some of its design features.
- The takeaways from the paper are clear and concrete, with actionable suggestions for developers and protocol designers.

## Reasons to reject
- While the root cause analysis identifies a number of causes of the performance gap, it doesn't really dig in to the relative impact of each. I'd assume that the UDP GRO issue that is identified is the main cause, but this isn't explicit in the paper.
- The paper mentions that the design of QUIC (and in particular, its ACK mechanisms) have an impact, but this isn't studied in depth.

**Questions:**

- Section 5.2 says "though TCP and QUIC have different ACK mechanisms, ..." before discussion the average packet RTT. How much of an impact do these different mechanisms have on the average packet RTT? If it is significant, then the difference between TCP and QUIC needs to be explained here.
- Section 5 ends with "for HTTP/2, the ACKS are handled by the OS kernel, and they are generated more efficiently and sparsely due to various optimizations such as TCP delayed ACK and receive offload" -- in a similar vein to the question above, how impactful are these "various optimisations"? And again, if this is significant, more needs to be said about them.

**Reviewer Confidence:**

3: The reviewer is confident but not certain that the evaluation is correct

**Scope:**

4: The work is relevant to the Web and to the track, and is of broad interest to the community

---

### Official Review · Reviewer_8Xne · 2023-11-27

**Novelty:** 2
**Technical Quality:** 6

**Review:**

As quic adoption continues to grow, there is a strong need to understand and address performance bottlenecks. To this end, paper which analyze the quic deployments are crucial.

The first point I want to make is that much of the performance issues are not because of the protocol but rather because of the deployment setup (e.g., offloading) and configuration (e.g., delayed-ACKs).  Given this point, the comments about delayed-ACKs should have been removed with the parameter tuning efforts (e.g., [1]). Additionally,  the trade-off between ACK frequency and CPU overheads has been highlighted by previous work (e.g., [2]).

The second point the behavior and implementation of QUIC varies highly across different clients. The current focus seems to be on a very narrow set of clients.   In particular, exploring a different client perhaps quiche would yield significantly different results --- perhaps delayed ACK would not be an issue given the extensive work done by cloudflare.


The third point to highlight is that the client/server were developed with similar limitations.  Much of the observations findings at the clients mirror those at the server side. This is unsurprising as the recommendations (ACK and offloads) are been made in many works as a way to optimize the servers or just the QUIC performance.

There is a clear need to understand and improve QUIC, this draft seems to analyze and focus on an aspect of QUIC deployment and configuration that a large body of work has focused on. Unfortunately the draft does not do a clear job of articulating the novelty. This is particularly troublesome as many of the recommendations are already quite well known.

[1] Reducing the acknowledgement frequency in IETF QUIC
https://aura.abdn.ac.uk/bitstream/handle/2164/19428/Castura_etal_IJSCN_Reducing_the_Acknowledgement_VOR.pdf;jsessionid=29C19C0AB869A4F137E4D6A17DD4B266?sequence=1

[2]QUIC on the Highway: Evaluating Performance on High-rate Links

**Questions:**

You current related work list papers without highlight how your analysis/observations differ. The key takeaways of delayed-ACK and offload have been (1) well analyzed, and (2) well known as recommendations. Given prior analysis and recommendations, how does this work inform the community?

**Reviewer Confidence:**

3: The reviewer is confident but not certain that the evaluation is correct

**Scope:**

4: The work is relevant to the Web and to the track, and is of broad interest to the community

---

### Decision · Program_Chairs · 2024-01-22

**Decision:**

Accept (Oral)

**Comment:**

The paper presents the findings of a thorough analysis of QUIC performance over fast Internet. The authors have made a setup of QUIC server (LSQUIC) and QUIC client under different bandwidth patterns and over different applications, ranging from the stand-alone applications to browser-based applications, as well as video streaming. Following robust good analysis, the authors conclude that QUIC does not perform well over fast Internet, and the reason behind that is the receiver-side processing overhead of user-space connection handling and congestion control mechanism on top of UDP.

 Overall, the analysis was generally well received, and the reviewers were all positive about the work.
 The reviewers mentioned that the methodology is well-detailed, and they specifically praised the root-cause analysis for its thoroughness. Overall, the feedback I read recognized that the work provides a new insight into QUIC performance by analyzing receiver-side processing overhead.

 This paper received the highest scores in my batch, and I thus recommend an accept.
 There are several suggestions that the reviewers have made in order to improve the quality of the work -- which I believe are feasible to be implemented for the camera ready (the authors also responded during the rebuttal phase). I believe that addressing the raised concerns and incorporating additional discussions will strengthen the paper and contribute further to the understanding of QUIC performance over fast Internet connections.